# SARS-CoV-2 Related Antibody-Dependent Enhancement Phenomena In Vitro and In Vivo

**DOI:** 10.3390/microorganisms11041015

**Published:** 2023-04-13

**Authors:** Emi E. Nakayama, Tatsuo Shioda

**Affiliations:** Research Institute for Microbial Diseases, Osaka University, Suita 565-0871, Japan; emien@biken.osaka-u.ac.jp

**Keywords:** ADE, enhanced respiratory disease, nucleocapsid, spike, SARS-CoV-2, COVID-19, anti-S, anti-N, cytokine, antibody

## Abstract

Antibody-dependent enhancement (ADE) is a phenomenon in which antibodies produced in the body after infection or vaccination may enhance subsequent viral infections in vitro and in vivo. Although rare, symptoms of viral diseases are also enhanced by ADE following infection or vaccination in vivo. This is thought to be due to the production of antibodies with low neutralizing activity that bind to the virus and facilitate viral entry, or antigen–antibody complexes that cause airway inflammation, or a predominance of T-helper 2 cells among the immune system cells which leads to excessive eosinophilic tissue infiltration. Notably, ADE of infection and ADE of disease are different phenomena that overlap. In this article, we will describe the three types of ADE: (1) Fc receptor (FcR)-dependent ADE of infection in macrophages, (2) FcR-independent ADE of infection in other cells, and (3) FcR-dependent ADE of cytokine production in macrophages. We will describe their relationship to vaccination and natural infection, and discuss the possible involvement of ADE phenomena in COVID-19 pathogenesis.

## 1. Fc Receptor (FcR)-Dependent Antibody-Dependent Enhancement (ADE) of Dengue Virus Infection

Historically, ADE has been an issue with dengue fever. There are four dengue virus serotypes, and it is known that about 1–5% of infected patients become severely ill when they are infected with a serotype that is different from that of their initial infection [1]. The emergence of critical dengue shock syndrome (DSS) and severe dengue hemorrhagic fever (DHF) was investigated in Thailand during the 1960s [2,3,4], and studies have shown that individuals with pre-existing immunity against dengue virus are more likely to develop severe DSS and DHF [5,6]. This is attributed to the presence of antibodies with low neutralizing activity, i.e., they have the ability to bind to the virus but do not inhibit infection and facilitate the entry of the virus into macrophages [7]. Macrophages express the Fc gamma receptor (FcγR) on the cell surface, which recognizes the Fc portion of immunoglobulin G [8]. Macrophages actively take up (phagocytose) and digest foreign substances coated by antibodies (called opsonization) through the binding between FcR and Fc [8]. Once weakly bound antibodies dissociate from the virion after entry into the cells, the macrophage becomes more susceptible to infection than in the absence of the antibody. If the amount of strongly binding neutralizing antibody is insufficient to cover the entire virion surface, Fc and FcR-mediated enhancement of virion uptake into macrophages causes augmentation of viral infection.

ADE of infection is a term used for in vitro experiments, while ADE of disease is used for in vivo experiments involving animals. Since dengue virus multiplies in macrophages, ADE of infection leads to an increase in the amount of virus in the body [9] and represents a clinical problem as ADE of disease (Figure 1 left). Multiple statistical analyses of a long-term pediatric cohort in Nicaragua showed that risk of severe dengue disease is the highest within a narrow range of preexisting suboptimal anti-dengue virus antibody titers levels, 1:21 to 1:80 [7]. In contrast, they observed protection from all symptomatic dengue infections at higher antibody titers. It was revealed that Sanofi’s dengue vaccine did not induce sufficient amounts of neutralizing antibodies against dengue virus type 2 among the four serotypes of dengue virus [10]. Moreover, in the Philippines, where the type 2 virus became prevalent, there was an increase in hospitalization cases among children vaccinated with the vaccine known as Dengvaxia [11]. As a consequence, the potential for ADE was one of the concerns in developing the SARS-CoV-2 vaccine.

## 2. Fc Receptor-Dependent ADE of SARS-CoV-2 Infection

An outbreak of pneumonia caused by severe acute respiratory syndrome coronavirus-2, SARS-CoV-2 emerged in 2019 [12]. The disease was named Coronavirus disease 2019, COVID-19, and includes microthrombi [13] of the lungs, lower limbs, hands, brain [14], heart, liver and kidneys [15]. Viruses of the family Coronaviridae (Coronaviruses) are classified into four coronavirus genera: alpha, beta, gamma, and delta. Severe acute respiratory syndrome virus (SARS), Middle East respiratory syndrome virus (MERS) and SARS-CoV-2 are beta coronaviruses. The coronavirus particle is an enveloped virus with a circular lipid bilayer of 100–160 nm in diameter. The surface of the particle is covered with S (spike) proteins. The internal genetic information is a single-stranded (+) RNA, which is the longest viral RNA at 30 kb and associates with the N protein to constitute a nucleocapsid [16]. The receptor of SARS-CoV-2 is angiotensin converting enzyme2 (ACE2) [17]. Although there is a possibility of infection of macrophages [18] and vascular endothelial cells [19], virus titers in blood are low [20,21], so certain immunological mechanisms instead of the direct effects of the virus are the probable cause of vasculitis and cytokine storm.

Zhou et al. cloned human antibodies against the SARS-CoV-2 S protein from COVID-19 patients and found 11 out of 48 (23%) significantly enhanced viral infection of Raji cells. Among the 11 enhancing antibodies, 9 were receptor binding domain (RBD)-binding antibodies, and 2 antibodies bound to the S1 region, but not RBD. Antibody-dependent viral entry was fully abrogated by mutations in the antibody FcR binding site, demonstrating the requirement of FcR binding for ADE of infection in B cells expressing FcγRII [22]. ADE of viral entry was not observed in a human erythroleukemic cell line, K562, which also expresses FcγRII and is commonly used for dengue virus ADE assays [23], suggesting that FcγRII expression in K562 cells is not sufficient for replication of SARS-CoV-2. Wang et al. also found an antibody, MW05, which enhanced SARS-CoV-2 pseudo-virus infection of Raji cells, but not of THP-1 or K562 cells [24]. Flow cytometry data revealed that Raji cells express relatively high levels of FcγRIIB, that THP-1 cells express high levels of FcγRIA and FcγRIIA, and that K562 cells express high levels of FcγRIIA, suggesting that high levels of FcγRIIB but not FcγRIIA are required for ADE of infection [25]. Although the detection of reporter gene expression was increased by ADE-inducing antibodies, no significant viral replication was observed in Raji cells [22].

Maemula et al. also showed the importance of FcγRIIA, but not FcγRIA or FcγRIII, in the presence of ACE2, while the expression of FcγRIIA alone did not mediate SARS-CoV-2 infection using a pseudo-typed virus and patient plasma. In the case of a combination of live virus, convalescent-phase plasma and macrophages, the virus genome levels were higher compared with those with control plasma. However, the expression levels of inflammatory cytokines were not changed by the ADE-inducing plasma relative to the controls. These results indicated that ADE-inducing antibodies may not contribute to abnormal cytokine production in macrophages [26].

Shimizu et al. attempted to infect macrophage-like cells differentiated from iPS cells used for ADE assays of dengue virus [27,28] with SARS-CoV-2; however, live virus replication was not achieved. On the other hand, the introduction of the ACE2 and transmembrane protease serine 2 (TMPRSS2) genes [17] resulted in cells becoming susceptible to viral infection and replication as well as IL-6 production [29,30], suggesting that ACE2 is essential for viral entry to the cytoplasm, the site of viral replication, by way of phagocytic vesicles or endosomes.

Taken together, probably due to the low expression of ACE2 or other unknown reasons, SARS-CoV-2 is not capable of productive infection in macrophages (Figure 1 right), in contrast to dengue virus (Figure 1 left). There are no reports indicating that the SARS or MERS viruses, which are closely related to SARS-CoV-2, replicate in macrophages [31,32]. Moreover, there are other reports of SARS-CoV-2 leading to non-productive infection in macrophages, halting in the middle of its replication cycle and resulting in an abortive infection [33,34,35]. Macrophages are highly phagocytic, so even if a staining signal for the viral proteins, including the N protein, is detected in the cells, it may only indicate that a phagocytosed virus has been detected. Therefore, if artificially FcR/ACE2 expressing cells or Raji cells are used to detect FcR-dependent ADE of infection in vitro, the results may not provide direct evidence that ADE of disease is occurring in vivo.

## 3. FcR-Independent ADE of SARS-CoV-2 Infection

The detailed protein structure of the SARS-CoV-2 S protein and enhancing antibody is a topic of great interest. Liu et al. reported that antibodies that recognize the specific “binding domain”, which is formed by the four amino acids W64, H66, V213, and R214 in the N terminal domain (NTD) of the SARS-CoV-2 S protein, can cross-link two S proteins, thereby changing the S position from “down” to “up” to facilitate the interaction between the S protein and ACE2 [36]. The existence of enhancing antibodies that induce structural changes in viral surface proteins and facilitate binding to receptors has been demonstrated for HIV [37]. In a one-step infection experiment using pseudo-virus infection, the number of indicator-positive cells increased five-fold when enhancing antibodies were present. However, in a multiple replication experiment using live virus, the titer was increased by two-fold at best in the presence of an enhancing antibody, while there was no difference in cells with higher susceptibility to infection, such as TMPRSS2-expressing VeroE6 cells [38] which are commonly used for virus isolation and propagation (unpublished data). One explanation for the difference between the two experiments using the pseudo-reporter virus and live virus is that an increase in the number of viruses entering one cell is not reflected in the number of progeny viruses emerging from that cell. Furthermore, when progeny viruses emerging from one cell spread to neighboring cells rather than infecting distant cells (cell-to-cell transmission), there is no room for antibody involvement. Similarly, Li et al. also isolated several monoclonal antibodies binding to NTD or RBD [39]. Selected RBD neutralizing antibodies also demonstrated enhancement of virus infection in vitro in an FcR-dependent manner, while five non-neutralizing NTD antibodies mediated FcR-independent enhancement of virus infection in vitro. Infection-enhancing antibodies show activity only in a limited number of antibody clones obtained from infected individuals. When there are sufficient amounts of neutralizing antibodies, the neutralizing antibodies can prevent S protein from interacting with the ACE2 receptor; thus, no infection enhancement is observed despite the enhancing antibodies changing the steric structure of the S protein [36].

The reason why Fc-independent ADE became a major public health concern was the expectation that vaccinated individuals would be more susceptible to infection than non-vaccinated individuals when a future mutant variant emerges that escapes from neutralizing antibodies by mutation of the RBD but not the NTD, thereby retaining only the binding of enhancing antibodies [40]. Fortunately, the delta variant (B.1.617.2), which was the major variant in the summer of 2021 [41,42,43], retained antigenicity similar to that of the alpha variant (B.1.1.7) [44], and the neutralizing antibodies produced by the vaccine provided good protection [45]. Omicron BA.1, which has been prevalent since the beginning of 2022 [46,47], has no mutations in the NTD four amino acids themselves; however, mutations of the surrounding amino acids, especially the mutation at 212 and deletion at 211, possibly affected the structure of the antibody binding site, and infection-enhancing antibodies might have reduced binding activity. In the case of BA.5, one of the four amino acids at position 213 was mutated from valine to glycine, and the V213 mutation has already been shown experimentally to reduce the binding of an enhancing antibody [36].

To assess the frequency of enhancing antibodies in the general population, Ismanto et al. compared over 64 million heavy chain antibody sequences from healthy unvaccinated subjects, healthy subjects vaccinated with an mRNA vaccine and COVID-19 patient repertoires with the 11 previously reported enhancing antibodies. They found that 17 out of 94 from COVID-19 patients and 9 out of 59 from healthy vaccinated, and only 2 out of 96 from healthy unvaccinated subjects bound to the enhancing epitope. It should be noted that some antibodies possessed higher binding affinity to the S protein from the delta variant, but most lost their ability to bind to the Omicron BA.1 variant [48].

## 4. ADE of Infection In Vivo

In vitro ADE of infection does not necessarily predict enhanced infection in vivo. Previous studies with vaccine-induced antibodies against SARS virus have shown in vitro enhancement of infection with no in vivo infection enhancement in hamsters [49]. Increased lung inflammation was reported to only rarely occur in macaques infused with SARS-CoV-2 enhancing antibody [39]. Three of 46 monkeys injected with enhancing monoclonal antibodies had higher lung inflammation scores compared with controls. Among the three, only one monkey had alveolar edema and elevated bronchoalveolar lavage inflammatory cytokines. One explanation for this discrepancy is that in vitro enhancing antibodies have the ability to suppress SARS-CoV-2 replication in vivo through certain FcR-mediated effector functions [50,51].

In vivo, non-neutralizing NTD antibodies not only mediate ADE, but may also mediate antibody-dependent cell-mediated cytotoxicity (ADCC), antibody-dependent cell mediated phagocytosis (ADCP), and complement dependent cytotoxicity (CDC). The immune system is activated after the virus enters the body. Opsonization itself can inactivate infectious virions. Enveloped viruses are also susceptible to lysis by the membrane-attack complex (MAC) of complements, so-called virolysis [52].

Indeed, studies on breakthrough infections among individuals who received a COVID-19 vaccine showed that most cases are asymptomatic or present as mild disease with few persistent infections [53,54], while case control studies showed that disease severity and hospitalization is lower in vaccinated versus unvaccinated individuals [55]. It was also pointed out that the low levels of virus-specific post vaccination neutralizing antibodies in patients with primary antibody deficiencies [56] may not protect from infection. However, humoral immunodeficiency itself may not be a risk of ADE of SARS-CoV-2 infection, since the ADE of infection is not observed in macrophages.

## 5. FcR-Dependent ADE of Cytokine Production

As described above, macrophages are not involved in the amplification of viral reproduction. At the beginning of the Wuhan strain outbreak, severe illness due to pneumonia was discussed as a sudden decrease in oxygen saturation about 5–7 days after disease onset [20,57,58]. Predictive markers of severe disease were investigated [21,59,60,61,62,63,64], and it was announced that high levels of TNF-α, IL-8 and IL-6 correlated with severe disease [63,65,66], leading to extensive clinical research on the usage of anti-IL-6R antibodies [67], JAK inhibitors [68] as well as steroids [69]. The severe pneumonia in COVID-19 is considered to be a form of macrophage activation syndrome [70]. This raises a question: how do macrophages produce cytokines without infection? To mimic the infected cells in the alveolar epithelium destroyed by SARS-CoV-2 and cellular immunity against it, researchers added crushed infected cells to macrophages and observed large amounts of IL-6 secretion, even though no productive infection was established [71]. Out of 24 open reading frame proteins, only the N protein, and S protein [72] induced IL-6. The N protein had a much higher ability to induce IL-6 secretion in macrophages than the S protein did. An antibody against the N protein was added to neutralize the IL-6 production; contrary to expectations, the antibody induced macrophages to produce more IL-6, and sera from patients with severe disease induced more IL-6 production compared with patients with milder disease [71]. It is known from earlier studies that severely infected patients have higher anti-N antibodies [73,74,75,76]. It was not clear at first whether this was a cause or a consequence; however, there is an explanation if we consider that N antibodies exacerbate the cytokine storm. It is known that older individuals are at higher risk of severe disease [77]; thus, it is reasonable that elderly people repeatedly exposed to closely related common cold coronaviruses might have memory B cells that can cross-react with the N protein of SARS-CoV-2 due to somatic hypermutation (see below). Indeed, existence of memory T cells that can cross-react with SARS-CoV-2 was shown in samples collected before 2019 [78]. The unclear therapeutic consequence of convalescent plasma therapy [79,80,81] also makes sense if we consider that the anti-N antibodies of recovered patients had a detrimental effect, i.e., they caused the hyperinflammation counteracting the protective effect of the neutralizing activity of anti-S antibodies.

The N protein promotes maturation of proinflammatory cytokines and induces proinflammatory responses in cultured cells and mice [82]. Mechanistically, the N protein interacts directly with the NOD-, LRR- and pyrin domain-containing protein 3 (NLRP3), and promotes assembly of inflammasomes [82]. More importantly, the N protein aggravates lung injury and promotes IL-1β and IL-6 activation in acute inflammation mouse models [82]. Moreover, tissue-resident macrophages, but not infected epithelial and endothelial cells, from lung autopsies of COVID-19 patients have activated inflammasomes [83,84]. Pyroptosis aborts viral infection before infectious virions are fully assembled [85], but the inflammatory mediators released from pyroptotic monocytes and macrophages can cause a cytokine storm (Figure 2).

When the N protein of the delta variant (which differs from wild-type Wuhan at D63G, R203M, and D377Y) and the N protein of omicron BA.1 (which differs at P13L, 31-33 ERS deletion and RG203-204KR) were expressed, IL-6 induction from iPS-derived myeloid cells was markedly reduced (Figure 3A), although the S protein itself mutated to be more and less fusogenic in the delta and omicron cultured cells, respectively. This is in good agreement with the fact that omicron BA.1 was less pathogenic compared with ancestral SARS-CoV-2 in a hamster model [86]. Most recently, Chen et al. introduced the S gene of omicron BA.1 into the ancestral virus and found that this chimeric virus still killed the hACE2 overexpressing mice (K18-hACE2), while the mice infected with omicron survived [87]. Furthermore, the additional introduction of the nsp6 protein from omicron resulted in very slow replication and low viral load in the mice, and good but not complete survival (20% mortality). Chen et al. concluded that the contribution of other proteins could not be completely ruled out since the mice died despite the low viral load and there being few antigen-positive cells among the epithelial cells of infected mice [87]. Nsp6 inhibits the lysosomal autophagy system by direct interaction with a lysosomal proton pump component and stimulates NLRP3-dependent cytokine production and pyroptosis in the lungs [88]; thus, it might be possible that mutations in the N protein may also contribute to the attenuated phenotype of the omicron variant.

## 6. Enhanced Respiratory Disease (ERD)

Formalin-inactivated vaccine against measles virus caused atypical measles in children [89]. Formalin-inactivated respiratory syncytial virus (FI-RSV) vaccination resulted in higher incidence of hospitalizations due to severe illness in children (80%) compared with non-immunized children (5%) [90]. The formation of virus–antibody complexes that activate immune cascades was thought to produce noticeable lung pathology [91]. Immunization of mice with FI-RSV elicited a T-helper cell type 2 (Th2) dominant response [92]. In mice challenged with RSV, enhanced disease along with lung inflammation and injury were found to be associated with pulmonary eosinophilia [93]. Priming with RSV G protein induced IL-4/IL-13 cytokines after RSV infection [94]. A similar Th2 response and enhanced infiltration of eosinophils in the lungs were observed in animals vaccinated with virus replicon particles expressing the N protein of SARS [95], inactivated whole virion of SARS [96], and a vaccinia virus vector expressing the SARS N protein [97] after virus challenge. Our proposed mechanism for ADE of cytokine production could explain why most of the previously developed vaccines against SARS failed to show protective immunity against the challenged virus. Furthermore, it is consistent with the fact that the efficacy of inactivated vaccines against severe disease is lower than that of mRNA-based vaccines [98,99,100,101,102,103].

In contrast, the effectiveness of the therapeutic monoclonal antibody treatment REGN-COV-02, a combination of imdevimab and casirivimab [104], against the delta variant (B.1.617.2) [105,106], and sotrovimab [107] against the omicron BA.1 variant infection [108] suggests that the antigen–antibody complex may not harm humans in SARS-CoV-2 infection.

## 7. Putative Involvement of ADE of Cytokine Production in COVID-19 Pathogenesis

The accumulated evidence during the COVID-19 pandemic indicates that SARS-CoV-2 infection causes micro-thrombosis that affects multiple organs, such as the heart, brain and kidneys, increasing the mortality burden in COVID-19 patients [109,110]. Figure 4 shows a schematic image of the involvement of ADE phenomena in COVID-19 pathogenesis. Macrophages in the lung and peripheral blood vessels, and microglia in the brain could play an important role in ADE of cytokine production when the co-existence of the N protein and anti-N antibody could amplify inflammation. Multiple organ failure is a downstream consequence of the cytokine storm. The resultant vasculitis can be thought of as micro-infarctions in the vascular beds of these organs. Hyperactivation of the complement system and tissue factor-enriched neutrophil extracellular traps (NETs) are key drivers in COVID-19 immunothrombosis [111]. Patients showed a higher number of polymorphonuclear neutrophils (PMN) forming NETs relative to healthy controls. The absolute number of PMNs forming NETs was inversely correlated with oxygen status and positively correlated with inflammatory markers such as C-reactive protein (CRP) and ferritin, and vascular cell adhesion molecule 1 [112]. Many more secondary pathological changes such as the formation of microclots and platelet activation have been observed in people with long COVID [113].

Complement activation is a critical event for COVID-19 disease [114]. A prospective study of 25 intensive care unit-hospitalized patients for up to 21 days revealed that severely ill COVID-19 patients had increased and persistent complement activation, mediated strongly via the alternative pathway [115]. Satyam et al. also found increased deposition of MAC (C5b-9) and a reduced deposition of complement factor H, a key inhibitor of the activation of the alternative pathway [116]. In addition, it is noteworthy that the complement cascade is hyperactivated via the lectin pathway by N protein in the lungs of COVID-19 patients [117,118]. The N protein bound to mannan-binding lectin-associated serine protease 2 and caused complement hyperactivation and inflammatory lung injury [119].

Local inflammatory responses to SARS-CoV-2 in one organ can cause lasting alterations in distant tissues and organs. SARS-CoV-2 infection of transgenic mice expressing human ACE2 exhibited a trend toward developing encephalitis rather than pneumonia, and Albornoz et al. observed the presence of the virus in the brain together with microglial activation and NLRP3 inflammasome upregulation in contrast to uninfected mice [120]. In a respiratory infection mouse model, there was a prolonged change in the central nervous system (CNS), including microglial activation, oligodendrocyte loss, and reduced myelination, despite the lack of evident symptoms/signs of illness. In this model, although no virus was detected in the brain, elevated cytokines, including IFN-γ, IL-6, TNF-α, and CXC/CC chemokines, were detected in the cerebrospinal fluid (CSF) [121].

Among Swedish adult COVID-19 patients exhibiting neurologic symptoms, viral antigen was detected in the CSF and correlated with CNS immune activation compared with control participants [122]. All the CSF samples were negative for SARS-CoV-2 RNA. In contrast, the SARS- CoV-2 N protein was detected in the CSF in 31 out of 35 patients, while only one patient was positive for the S protein. Furthermore, the CSF N protein levels were correlated with the CSF immune activation biomarkers IFN-γ and neopterin. Patients with signs of neuroaxonal injury had more elevated inflammatory cytokines that were not attributable to differences in pneumonia severity. These results suggest that the viral component N protein can contribute to CNS immune responses without direct invasion of the virus into the CNS [122]. It should be noted that S protein is membrane anchored due to the endoplasmic reticulum retention signal located in its cytoplasmic tail [123], while the N protein is expressed in the cytoplasm and is released from infected cells following apoptosis or cytotoxic T lymphocyte-induced cell death. While the existence of anti-N antibodies in the brain has not been confirmed yet, severely ill patients who died during the acute phase exhibited multifocal vascular damage, as determined by leakage of the serum proteins IgG and IgM into the brain parenchyma [124] with endothelial cell activation. Platelet aggregates and microthrombi adhered to endothelial cells along the vascular lumina. Immune complexes with activation of the classical complement pathways C1q and C4d were found on endothelial cells and platelets. Perivascular infiltrates predominantly consisted of macrophages and some CD8+ T cells. Although the target protein of IgG and IgM is unclear, antibody-mediated cytotoxicity directed against endothelial cells seems to trigger platelet aggregation, vascular leakage, neuroinflammation and, hence, neuronal injury [124]. Moreover, IL-6 over-stimulation is likely to be involved in mediating blood–brain barrier leakage [125].

## 8. ADE of Disease as a Concern of Re-Infection

In Manaus, the capital of the Brazilian state Amazonas, SARS-CoV-2 spread rapidly and nearly 70% of the population possessed anti-N antibody by October 2020 [126]. The evolution of a new lineage P.1, which was named the gamma variant, later emerged in Manaus [127], indicating that herd immunity may not have been achieved despite the high attack rate. The lineage B.1.351, first reported in South Africa and later renamed as the beta variant [128], and the omicron variant, originally B.1.1.529, which were reclassified into BA lineages, emerged in South Africa [129], where the prevalence of immunocompromised hosts due to HIV infection can be high. More recently, cases of individuals previously infected with BA.1 and subsequently infected with omicron BA.2 were reported, despite the nearly 70 to 90% protective efficiency [130,131], which were attributed to the rapid antibody titer decay. The effectiveness of infection with pre-omicron strains against symptomatic BA.4 or BA.5 reinfection was 35.5% [132] and BA.5 re-infection after BA.1 infection is common. Re-infection will be a problem since new emerging variants often gain mutations to escape neutralizing antibodies.

Why do antibodies against SARS-CoV-2 decay so quickly? Kaneko et al. showed that there is a marked loss of germinal centers in lymph nodes and the spleen, depletion of Bcl-6+ B cells but preservation of Tbet+ B cells, and aberrant extra-follicular TNF-α accumulation in acute COVID-19 patients. These results might explain the limited durability of antibody responses and suggest the difficulty of herd immunity achievement through natural infection [133].

It is worth noting that children have higher antiviral sensing ability and a stronger antiviral interferon response in the upper airways since they have higher levels of basal expression of the viral pattern recognition receptors MDA5 and RIG-I and gene expression signatures associated with IFN-α signaling in nasal epithelial cells, macrophages and dendritic cells (DC) [134,135]. This pre-activated innate immune system may more efficiently eliminate SARS-CoV-2 infection in children. Indeed, analysis of the three children and their parents with PCR-confirmed symptomatic SARS-CoV-2 infection suggested that children may establish an effective early antiviral immune response to clear the virus without any detected PCR evidence of SARS-CoV-2 infection [136]. This strong innate immunological response affects the acquisition of memory T cells. A longitudinal multimodal analysis showed that SARS-CoV-2 produces only small changes in the circulating T cell population in children with mild/asymptomatic SARS-CoV-2 infection compared with their parents with the same disease severity who had more evidence of systemic T cell activation. The children possessed diverse polyclonal SARS-CoV-2-specific naïve T cells, whereas the adults showed clonally expanded SARS-CoV-2-specific memory T cells [136]. This was associated with the development of robust CD4+ memory T cell responses in adults but not in children [137]. These data suggest that rapid clearance of SARS-CoV-2 by innate immune responses in children may weaken their acquired immunity and ability to resist re-infection. Furthermore, it demonstrated that COVID-19 patients showed a deficit in some DC subsets and alterations in homing and activation markers of DC, which are not recovered more than 7 months after infection, regardless of previous hospitalization [138]. These findings suggest the possibility that children, who depend more on innate immunity with plasmacytoid DC function than adults, will acquire more damage to their resistance against subsequent infection, not only for secondary infection of SARS-CoV-2 but also for bacterial and other viral infections after COVID-19.

## 9. Is Vaccination Required for Vaccine Naïve but Infected Individuals to Establish Hybrid Immunity?

There has been some disagreement on whether vaccination is necessary for SARS-CoV-2 infected and recovered individuals [139]. The symptoms and problems are not limited to pneumonia but extend to dysfunction of other organs such as the heart and brain, and the impacts of long-COVID should be considered.

The US Department of Veterans Affairs national healthcare database was used to build a cohort of 443,588 individuals with a single SARS-CoV-2 infection, 40,947 with re-infection (two or more times) and 5,334,729 uninfected controls [140]. Compared with single infection, re-infection contributed additional risks of death (hazard ratio (HR) = 2.17) and hospitalization (HR = 3.32). The risks were evident in the acute phase but persisted in the post-acute phase at 6 months, with symptoms including pulmonary, cardiovascular, hematological, diabetes, gastrointestinal, kidney, mental health, musculoskeletal and neurological disorders [140]. It is not clear whether the secondary infection was more severe than the primary infection; however, if only 1% of re-infected cases become severe, this may result in a large burden on health services since the target population size is extremely large. Thus, continued vigilance to reduce the risk of re-infection may be important for people who have been previously infected.

The level of N protein is the benchmark factor that is considered when predicting disease severity [141]. Plasma N protein levels of 1000 ng/L or greater are associated with markedly higher risks of worsened pulmonary status at the acute phase (odds ratio = 5.06) and longer time required for hospital discharge (median, 7 vs. 4 days). Moreover, plasma N protein levels were higher in those who lacked anti-S antibodies [141]. Therefore, the neutralizing titer of anti-S antibody is critical for reducing levels of the N protein in vivo as well as the virus itself and the S protein, which are all involved in the pathogenesis of COVID-19.

There is an argument for vaccination with the disappeared wild Wuhan strain or the disappearing omicron BA.1 or BA.5 variants being sufficient to induce protective immunity against newly emerging omicron variants and future mutated viruses. However, vaccines composed of inactivated whole virion including the N protein are not recommended, as discussed in this review. It should be noted here that the antibody titers induced by in-activated whole virion were lower than for other vaccines [101], especially in the levels of IgA in the nasal epithelial lining fluid [142].

The four human IgG subclasses, IgG1 to IgG4, have distinct effector properties due to differences in FcR binding and complement activation. The human IgG4 normally exists in the serum at lower concentrations than IgG1, IgG2, or IgG3. A longitudinal analysis of the level of anti-S antibodies from each IgG subclass in recipients of the SARS-CoV-2 mRNA vaccine [143] revealed that IgG4 antibodies among all S-specific IgG antibodies increased from 0.04% shortly after the second vaccination to 19.27% after the third vaccination. Serum antibody effector activity, as assessed by antibody-dependent phagocytosis or complement deposition, was reduced after the third dose compared with after the second dose [143]. It should be noted that a third vaccination elicited superior neutralizing immunity to all the variants of concern [144]. Furthermore, the avidity was significantly increased after the third vaccination and superior neutralization of pseudo-typed virus was observed. In this study, antibody-mediated phagocytic activity and complement deposition related to ADE of disease were reduced in sera after the third vaccination. Because Fc-mediated effector function could be important for viral clearance, an increase in IgG4 subclasses might result in longer viral persistence in patients. However, it is also true that noninflammatory Fc-mediated effector functions reduce immunopathology, whereas viruses are efficiently neutralized via high-avidity antibody variable regions of IgG4 antibodies. In a cohort of vaccinees with breakthrough infections, no evidence of alteration in disease severity was found [143]. In addition, adverse effects related to antigen–antibody immune complex formation have not been observed in recipients of AstraZeneca’s Evusheld (AZD7442; tixagevimab and cilgavimab), in which the L234F/L235E/P331S modification reduces binding to FcγR and the C1q-like IgG4 isotype [145], indicating that the immunoglobulin class switch of IgG4 may control the balance between binding maturation and over-stimulation of inflammation without severe negative effect. Regardless of the immunoglobulin class switch, the somatic hypermutation and affinity maturation of the epitope binding region of antibody genes in memory B lymphocytes is important [146,147,148,149,150,151]. A broadly neutralizing antibody that has potent cross-reactivity against a wide variety of variants was reported in HIV infected individuals [152]. A long exposure interval is required to generate cross-neutralizing potency against different variants antigenically distant from the ancestral strain [153]. The ratio between protective anti-S antibody and detrimental anti-N antibody in each individual [154] may affect the clinical relevance with breakthrough infection or re-infection.

## 10. Management of ADE Caused by the N Protein and Antibody

Kang et al. found that nCoV396 antibody recognizes the N protein of SARS-CoV-2 as well as those of the SARS and MERS viruses. The 162–170 region of the SARS-CoV-2 N protein is an epitope since three amino acids (Q163, L167, and K169) are conserved among these three viruses. Structural analysis revealed that nCoV396 antibody binding of the SARS-CoV-2 N-terminal domain of the N protein (N-NTD) undergoes several conformational changes, resulting in an enlargement of the N-NTD RNA binding pocket and partial unfolding of the basic palm region [155]. This conformational change occurs in the C-terminal portion of N-NTD, which may alter the positioning of individual domains in the full-length protein, leading to a potential allosteric suppressive effect on N protein-induced complement hyperactivation [155]. On the other hand, Sen et al. reported the levels of antibody against a specific epitope of the N protein can effectively predict severe disease (specificity = 83.6%). The presence of this antibody was correlated with high levels of IL-6 [156]. Surprisingly, this Ep9 peptide consists of residues 152 to 172, which overlaps with nCoV396 recognition. In another report from Singapore, there were correlations between antibody responses against the N4P5 epitope and pneumonia and the tissue damage markers CRP and lactate dehydrogenase [157]. The N4P5 epitope, residues 153 to 170, is very similar to Ep9. These reports prompted us to mask the epitope in the NTD of the N protein with a monoclonal antibody lacking binding activity to FcR to block the IL-6 enhancing effect in patient serum. As shown in Figure 3B, IL-6 levels produced from cells in the presence of N protein in patients’ sera were decreased in a dose-dependent manner by the addition of a monoclonal antibody.

While C5 inhibition has been discussed from a therapeutic perspective [158], significant improvements have not been observed to date and must be interpreted in light of concerns about bacterial infection. NLRP3 signal inhibitors have also been proposed [82]. Small molecule compounds that specifically target the interaction between the N protein and the NLRP3 cascade would be safe drugs for abrogating the cytokine storm.

## 11. Why Focus on ADE?

Over 18,000 cases of measles were reported in the Philippines in 2018, compared with about 2400 in the previous year. The measles vaccination rate fell from 88% in 2014 to 73% in 2017, and then to about 55% in 2018. The sharp drop came in the wake of a political battle over Sanofi’s dengue vaccine, Dengvaxia, which was discontinued in the Philippines over safety concerns regarding ADE [159]. This situation should be avoided when the next pandemic arrives, which is expected soon after the COVID-19 pandemic ends.

Vaccination is an essential tool in the control of infectious diseases, and any detrimental effects on vaccine recipients are carefully monitored. Since vaccination targets the predominantly healthy population, ADE phenomenon must always be evaluated in the development of new types of vaccines. In addition, the emergence of further SARS-CoV-2 variants capable of escaping neutralization is also a public health concern. The development of cutting-edge protease inhibitors to control the COVID-19 pandemic and long COVID [160,161] is highly anticipated.

## Figures and Tables

**Figure 1 microorganisms-11-01015-f001:**
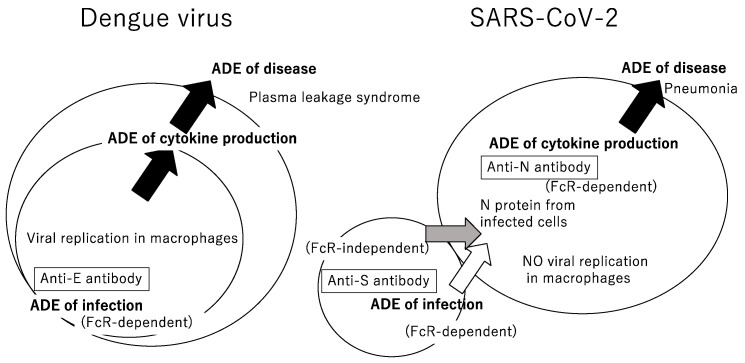
Schematic representation of antibody-dependent enhancement (ADE) of infection and disease in dengue and COVID-19. (**Left**) In the case of dengue, enhanced viral infection to peripheral blood monocytes and macrophages via the Fc receptor (FcR) is a direct consequence of an increase in viral load in the blood. The infected macrophages produce cytokines such as IL-6 and IL-8, and subsequent plasma discharge, hemo-concentration, and loss of platelets trigger severe dengue disease, hemorrhagic fever, and shock syndrome. (**Right**) In the case of COVID-19, SARS-CoV-2 replication in macrophages is not observed, while the entry of N protein released from infected cells proximal to macrophages is assisted by anti-N antibodies in an FcR-dependent manner, resulting in increased secretion of cytokines from macrophages, cytokine storm, and severe disease.

**Figure 2 microorganisms-11-01015-f002:**
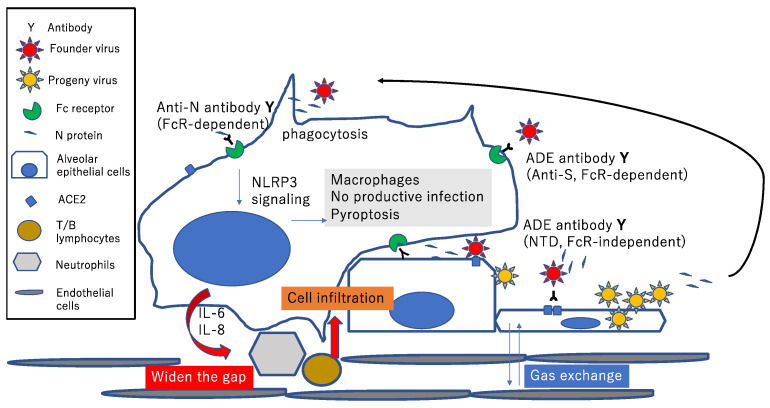
Potential pathways contributing to hyperactivation of macrophages and hyperinflammation in the alveolar cavity of COVID-19 patients. The right side shows the anti-S antibody-mediated, FcR-dependent, and FcR-independent enhancement of infection. The N-terminal domain (NTD) of the S protein is the main target of FcR-independent ADE antibody. Types I and II alveolar epithelial cells are infected through the ACE2 receptor, while alveolar macrophages do not support productive infection. The progeny virus and soluble N protein are released from infected epithelial cells. The N protein is recognized by the anti-N antibody. The left side shows FcR-dependent ADE of cytokine production. The C-terminal region of the N protein stimulates the inflammasome via NOD-, LRR-, and pyrin domain-containing protein 3 (NLRP3), and IL-6, IL-8 and TNF-α are secreted. IL-8 attracts neutrophils, which form neutrophil extracellular traps, damaging tissues. In addition, plasma leakage in the lung also reduces oxygenation.

**Figure 3 microorganisms-11-01015-f003:**
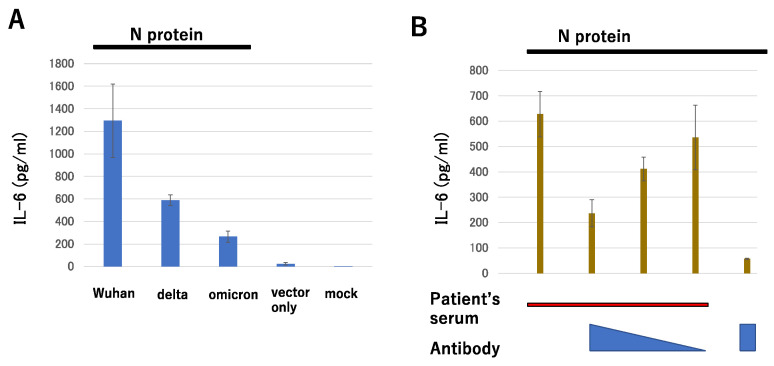
SARS-CoV-2 N protein induced IL-6 production. IL-6 levels were measured by ELISA 2 days after treatment. (**A**) iPS-derived myeloid-like K-ML2 cells were stimulated with the lysate of 293T cells transfected with plasmids encoding each of the SARS-CoV-2 N proteins. (**B**) K-ML2 cells were incubated with a patient’s serum in the presence or absence of serially diluted anti-N monoclonal antibody lacking binding activity to human FcR together with 156 ng/mL of N protein.

**Figure 4 microorganisms-11-01015-f004:**
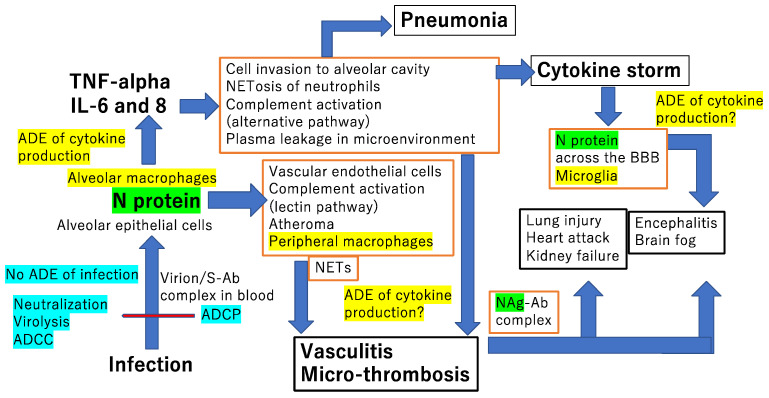
Putative involvement of ADE phenomena in COVID-19 pathogenesis. The bold indicates the anti-N antibody-dependent enhancement of cytokine production shown by Nakayama et al. [71]. When the macrophage lineage cells (marked in yellow) in tissues or blood come into contact with anti-N antibody and N protein (marked in green), there is ADE of cytokine production from macrophage lineage cells in the lungs, peripheral blood vessels, and the brain. The blue color indicates the protective effect of anti-S antibody including antibody dependent cell-mediated cytotoxicity (ADCC) related to natural killer cells and antibody dependent cell-mediated phagocytosis (ADCP). Cytotoxic T lymphocytes, which are a main contributor to the elimination of infected cells, are not mentioned in the figure. The red boxes indicate the contributing factors in the microenvironments of the lungs, blood vessels and brain, whereas the black boxes show the clinical symptoms. NETosis denotes the death of neutrophils with neutrophil extracellular traps (NETs) formation.

## Data Availability

All data and materials used and/or analyzed during this study are included in this published article.

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
