# Peer review of "SARS-CoV-2 Related Antibody-Dependent Enhancement Phenomena In Vitro and In Vivo"

_microorganisms, 2023, doi:10.3390/microorganisms11041015_

Round 1

Reviewer 1 Report

The abstract needs adjustments such as inserting information about the authors' conclusions. Line 26 – Although studies show that previous infection with DENV can increase ADE, it is important to emphasize that this occurs when the first infection occurs less than 1 year after the second infection. Line 396 – Despite the decrease in antibodies, it is important to emphasize that another reason for the reinfection of patients was the generic variability between the variants. I suggest that this be addressed. Line 412 – It was not clear how the difference in action between children and adults occurs. Line 459 – Vaccines produced from inactive virions are less efficient than other vaccine technologies. However, it does not seem plausible to me that these vaccines cannot be used as the text suggests. Therefore, I suggest a review of this conclusion. The work is very interesting with several pertinent points for the reader's reflection on ADE and its relationship with Dengue and SARs-CoV-2. However, the author's concern with the vaccination process, especially in relation to the different vaccine technologies used with SARS-CoV-2, left a little to be desired. Because we know that different vaccines produce different rates of antibody production. Therefore, I suggest that the author compare the level of antibody production between SARS-CoV-2 vaccines so that his arguments gain more strength for the conclusions of the article.

Reviewer 2 Report

Nakayama et al. provide a comprehensive insight into the issue of ADE phenomena in their manuscript. The manuscript is very well written.

Here, I provide a few comments and queries that could be reflected in the revised version of the manuscript:

1) I recommend adding other examples of ADE such as RSV or measles (e.g. Munoz et al., doi: 10.1016/j.vaccine.2021.01.055)

2) ADE may be enhanced by an insufficient level of virus-specific neutralizing antibodies. Low levels of virus-specific postvaccination neutralizing antibodies have been described in patients with primary antibody deficiencies (e.g. Milota et al., doi: 10.1016/j.jaip.2022.10.046). Are the patients with primary antibody deficiency or other immunocompromised patients at higher risk of ADE? Please, discuss!
